

# Impacts of long-range transported mineral dust on summertime convective cloud and precipitation: a case study over the Taiwan region

Yanda Zhang[1], Fangqun Yu[1], Gan Luo[1], Jiwen Fan[2], and Shuai Liu[3]

[1]Atmospheric Sciences Research Center, State University of New York, Albany, NY 12203, USA
[2]Atmospheric Sciences and Global Change Division, Pacific Northwest National Laboratory, Richland, WA 99354, USA
[3]Institute of Atmospheric Physics, Chinese Academy of Sciences, Beijing 100029, China

*Correspondence to*: Yanda Zhang (yzhang31@albany.edu) and Fangqun Yu (fyu@albany.edu)

**Abstract.** As one of the most abundant atmospheric aerosols and effective ice nuclei, mineral dust particles affect clouds and precipitation in the Earth system. Here numerical experiments are carried out to investigate the impacts of dust aerosols on summertime convective clouds and precipitation over the mountainous region in Taiwan. We run the Weather Research and Forecasting model (WRF) with the Morrison two-moment and spectral-bin microphysics (SBM) schemes at 3 km resolution, with the dust number concentrations from a global chemical transport model (GEOS-Chem-APM). The case study indicates that the long-range transported mineral dust, with relatively low number concentrations, can notably affect the properties of convective cloud (ice/liquid water contents, cloud top height, and cloud coverage) and precipitation (spatial pattern and intensity). The dust effects are evident during the strong convective periods, significantly increasing the ice water contents in the mixed-phase regime via the enhanced heterogeneous freezing. With both the Morr2 and SBM schemes, we see invigoration effects of dust aerosols on the convective intensity through enhanced condensation and deposition latent heating. In this process, the low-altitude dust particles are uplifted to the freezing level by updrafts which, in turn, enhance the convective cloud development through immersion freezing and convective invigoration. Comparing to the Morr2 scheme, the SBM scheme predicts more realistic precipitation and different invigoration effects of dust. The differences are partially attributed to the saturation adjustment approach utilized in the bulk schemes, which leads to the stronger enhancement of condensation at mid-low altitude and weaker deposition increase at the upper level.

## 1 Introduction

For the enormous global emission rate (Zender et al., 2004; Textor et al., 2006) and the long-range transport ability (Husar et al., 2004; Perry et al., 2004; Engelstaedter et al., 2006; Liu et al., 2008; Uno et al., 2009), mineral dust is one of the most abundant aerosol components in the atmosphere (Andreae et al., 1986; Carslaw et al., 2010; Kok et al., 2017). Previous studies indicate that mineral dust significantly affects the air quality (Prospero, 1999), public health (Pope et al., 2002; Chiu et al., 2008), biogeochemical cycles (Jickells et al., 2005), and climate systems. In the atmosphere, mineral dust is suggested to have



important impacts on the radiation budget of the Earth system through direct effects (scattering and absorbing shortwave and longwave radiations) (Li et al., 1996; Miller and Tegen, 1998; Zhao et al., 2011; Guo and Yin, 2015), semi-direct effects through changes in atmospheric temperature structure and cloud lifetime (Hansen et al., 1997; Ackerman et al., 2000; Koren et al., 2004; Huang et al., 2006).

    Besides the dust effects on the radiative properties of the atmosphere, dust aerosols are also suggested to influence the

cloud properties, enhance or suppress the clouds and precipitations by serving as cloud condensation nuclei (CCN) and ice nuclei (IN) (Gunn and Phillips, 1957; Rosenfeld, 2000; Yin et al., 2002; Sassen et al., 2003; Chen et al., 2008; Han et al., 2009; Min et al., 2009; Solomos et al., 2011; Seigel et al., 2013; Creamean et al., 2015; Creamean et al., 2016; Zhang et al., 2020b). For its large atmospheric loading and good ice-nucleating ability, mineral dust is well recognized as the most important atmospheric ice nucleating particle (INP) (Knopf and Koop, 2006; Heymsfield et al. 2007; Klein et al., 2010; Si et al., 2018),

and a series of field observations and laboratory studies demonstrate the dust effects on the ice-nucleating and heterogeneous freezing in the cirrus and the mixed-phase clouds (Roberts and Hallett, 1968; Zimmermann et al., 2008; Hoose and Möhler 2012; Atkinson et al., 2013; DeMott et al., 2010; DeMott et al., 2015). However, the dust effects on cloud and precipitation remain to have significant uncertainties and are highly dependent on the specific cloud types and meteorological conditions (Huang et al., 2014). Near the dust sources, the outbreaks of dust events provide high number concentrations of INPs, leading

to more ice particles at warmer temperatures through heterogeneous ice nucleation processes (Bangert et al., 2012), which can influence the mixed-phase clouds and enhance the cloud water and precipitation (Smoydzin et al., 2012; Min et al., 2014; Gibbons et al., 2018). Meanwhile, in the relatively dry atmosphere, water vapor competition limits ice particles' growth and reduces the effective radius of cloud particles, suppressing the precipitation as a result (Rosenfeld et al., 2001; Levin et al., 2005; Teller and Levin, 2006; Min et al., 2009). Over regions remote from dust sources, the transport of dust at the middle and

upper-troposphere are indicated to lead to dust entrainment into the mixed-phase clouds , contributing to ice nuclei populations and enhancing the precipitation under the condition with adequate water vapor inputs (DeMott et al., 2003; Muhlbauer and Lohmann, 2009; Ault et al., 2011; Creamean et al., 2013; Fan et al., 2014, 2017).

    In this paper, we choose the Taiwan region to study the impacts of the mineral dust on deep convective clouds and precipitation. Over the Taiwan area, the atmospheric dust loading is dominated by the long-range transported dust from various

dust sources (East Aisa, Sahara, Middle East, , etc.) (Hsu et al., 2012; Lin et al., 2012; Chen et al., 2015; Chou et al., 2017). Taiwan is also significantly affected by the summertime extreme rainfall events (Chen and Chen 2003; Xu et al., 2009; Chen et al. 2010; Lin et al., 2011; Yim et al., 2015; Zhang et al., 2020a). In our previous studies (Zhang et al., 2020b), we found a positive correlation between the number concentration of dust particles and the summertime rainfall intensity over Taiwan, showing that the cloud water and non-typhoon rainfall generally increase with the atmospheric dust loading over the mountain

region. The main objective of this paper is to investigate the detailed physical processes underlying this positive correlation, focusing on the impacts of the low-altitude transported mineral dust (under 4 km) which has relatively low number concentrations ($< 1$ cm$^{-3}$). In this study, we incorporate the DeMott et al. (2015) immersion freezing parameterization into cloud microphysics schemes in the Weather Research and Forecasting (WRF) model (Skamarock et al., 2019) to connect dust



with ice nucleation, following Fan et al. (2014). We utilize this modified WRF model to simulate a heavy rainfall case,
occurring on 08 July 2006, influenced by long-range transport of dust over the mountain region in Taiwan.

## 2 Data and Methodology

Influenced by a remote typhoon system, the circulation provided a strong water vapor transport to the mountainous region
at the north-western part of Taiwan island, leading to a severe precipitation event on local time (LT) 08 July 2006, with the
24-h accumulated precipitation over 220 mm in the precipitation center. At the same time, a long-range transported dust event
occurred over this region during the period (Fig. 1a). Here we use numerical experiments and multiple datasets to study the
interactions between dust aerosol and the convective cloud and precipitation.

### 2.1 Data

The datasets used in this study are listed below:

(1) Dust number concentrations. In this study, the DeMott et al. (2015) immersion freezing scheme is used to describe
the INP formation rate which is dependent on the atmospheric temperature and dust number concentration (diameter > 500
nm) ($N_d$). For the lack of measurements, $N_d$ simulated by the GEOS-Chem model with size-resolved advanced particle
microphysics (APM) (Yu and Luo, 2009) is used as the dust conditions in the WRF simulations. We run the GEOS-Chem-
APM model at 2°×2.5° horizontal resolution with 47 vertical levels, and the daily and hourly $N_d$ simulations are globally
outputted at selected sites. The GEOS-Chem-APM model is driven by meteorological input from the Goddard Earth Observing
System (GEOS) of the NASA Global Modeling and Assimilation Office (GMAO). In this model, dust emission is calculated
with the Mineral Dust Entrainment and Deposition (DEAD) scheme (Zender et al., 2003), and dust aerosols from 30 nm to 25
μm are represented by 15 bins. The GEOS-Chem dust simulations have been widely used in studies of dust aerosol (Generoso
et al., 2008; Kim et al., 2015; Ridley et al., 2016; Xu et al., 2017) and the GEOS-Chem dust simulation over the Taiwan region
is validated by our previous research (Zhang et al., 2019; Zhang et al., 2020a).
(2) MERRA-2 dust mixing ratio reanalysis. The time series and vertical profile of dust aerosols can be difficult to
continuously observe, especially during summertime with relatively low dust loadings. For the validation of the GEOS-Chem
dust simulation, the MERRA-2 reanalysis of dust mixing ratio (inst3_3d_aer_Nv) (Gelaro et al., 2017; GMAO, 2015) is used
as a reference for the time series and vertical distributions of the atmospheric dust aerosols. The MERRA-2 is a NASA
atmospheric reanalysis for the satellite era using the Goddard Earth Observing System Model, Version 5 (GEOS-5) with its
Atmospheric Data Assimilation System (ADAS), version 5.12.4. The MERRA project focuses on historical climate analyses
for a broad range of weather and climate time scales and places the NASA EOS suite of observations in a climate context. The
three-dimensional aerosol reanalysis (inst3_3d_aer_Nv) is 3 hourly data at 0.5°×0.625° horizontal resolution with 72 vertical
layers. As the dust number concentration is not directly provided in the MERRA-2 reanalysis, the MERRA-2 dust mixing ratio





in July 2006 is used to compare with and validate the temporal and vertical distributions of GEOS-Chem-APM dust mass
concentration simulation.

(3) Precipitation measurements. The daily precipitation data with1-km resolution from the Taiwan Climate Change
Projection Information and Adaptation Knowledge Platform (TCCIP) project is used as the rainfall observation. The TCCIP
precipitation data is widely used in previous studies on the extreme rainfall events in Taiwan (Chen and Chen, 2002; Chen et
al., 2007; Su et al., 2012; Lin et al., 2015; Kuo et al., 2016; Henny et al., 2021). We also use the half-hourly rainfall observation
at 0.1°×0.1° horizontal resolution from the Integrated Multi-satellitE Retrievals for GPM (IMERG) (Huffman et al., 2019) as
the rain rate observation.

(4) Satellite observation. In this study, the cloud measurements from the Moderate Resolution Imaging Spectroradiometer
(MODIS) aboard the Aqua (EOS PM) satellites (Platnick et al., 2015) are used to compare with the model simulations. The
level-2 MODIS/Aqua (MYD06_L2) cloud product consists of cloud optical and physical parameters at a spatial resolution of
either 1 km or 5 km, which are derived using remotely sensed infrared (cloud top temperature, cloud top height, effective
emissivity, cloud phase and cloud fraction), visible (cloud optical thickness and effective particle radius and cloud shadow
effects) and near-infrared (additional information in the retrieval of cloud particle phase) solar reflected radiances.

In this study, all the data are processed into local time (LT) (Greenwich Mean Time +8).

## 2.2 Model configuration and experiment design

### 2.2.1 Immersion freezing parameterization and dust aerosol

To study the detailed physical processes of the dust-cloud-precipitation interactions in this case, we conducted numerical
experiments with the WRFv4.20 Model. As given in Fig.1b, two nested domains with horizontal grid spacings of 9 km
(181×181) and 3 km (223×223) for Domain 1 and Domain 2, respectively, are used with 50 vertical levels up to 50 hPa. Hourly
ERA5 reanalysis is used as the model initial and boundary meteorological conditions. The physical parameterization schemes
used in the model include the RRTMG longwave radiation scheme, the RRTMG shortwave radiation scheme (Iacono et al.,
2008), the Mellor-Yamada-Janjic PBL scheme (Janjic 1994). Tiedtke cumulus scheme (Tiedtke 1989) is used in D01 and
turned off in D02. The simulations were initiated at 0000 UTC 07 July with 16 hours spin-up time and ended at 0000 UTC 09
July 2006. Simulations in D01 are mainly used to provide meteorological conditions for D02, and the analysis in this study is
focused on the fine-resolution simulation results in D02.

In the summertime, the dust number concentration is relatively low over this region remote from dust sources, with the
$N_d \approx 0.12$ cm$^{-3}$ in the top 10% of dust events simulated by the GEOS-Chem model (Zhang et al., 2020a). During this heavy
rainfall event, the atmospheric dust loading over this region is too low to influence the optical properties of the atmosphere
(~0.6 cm$^{-3}$), the direct effects of dust on solar radiation are not considered and this study focuses on the effects of dust as INPs.
To investigate the impacts of dust aerosols on cloud and precipitation, we employed two cloud microphysics parameterizations:
the spectral-bin microphysics scheme (SBM; Khain et al. 2004; Fan et al. 2012) and two-moment Morrison microphysics



scheme (Morr2; Morrison et al., 2009). This is to evaluate how the dust effects vary with the two typical types of cloud microphysics parameterizations. Based on the SBM and Morr2 schemes, we designed six sensitivity experiments using different settings of the immersion freezing parameterizations and dust conditions (Table 1). In the control runs (Morr2-Org and SBM-Org), the default Morr2 and SBM schemes in WRF v4.2.0 with Bigg (1953) immersion freezing parameterization are utilized without consideration of the dust effect. In the four dust-related runs (Morr2-Clean & Dusty, SBM-Clean & Dusty), to connect dust with ice nucleation, the immersion freezing parameterization of DeMott et al. (2015) is implemented to replace the original Bigg (1953) immersion freezing in both SBM and Morr2 microphysics schemes. In the Clean and Dusty runs, two vertical profiles of $N_d$ from the GEOS-Chem-APM model are used as the initial and boundary conditions of WRF simulations to represent the atmospheric dust number concentration in the clean and dusty conditions.   The $N_d$ vertical profile simulated by the GEOS-Chem-APM model during this event (07–08 July 2006) is used in the Morr2-Dusty and SBM-Dusty runs, to represent the $N_d$ condition of this long-range dust transport event (red dots in Fig. 1c); and for comparison to investigate the dust effects, the long-term mean $N_d$ profile averaged in the clean atmosphere (with $N_d$ < the 50$^{th}$ percentile of daily dust number concentrations in July) is used in the clean runs (Morr2-Clean and SBM-Clean) (blue triangles in Fig. 1c). The vertical profiles of $N_d$ show that, during this long-range dust transport event, the dust number concentration increased by about two orders of magnitude from the clean atmosphere. This mineral dust transport event occurred at a low altitude (under ~3 km) with no evident impacts on $N_d$ at the upper level (above ~6 km). Following Fan et al. (2014), the horizontal and vertical transports of dust are driven by WRF dynamic and ice formation through immersion freezing acting as the sink of dust and liquid droplets. The rainout and dry deposition of dust are not considered in this study. Here only the immersion freezing is modified to depend on dust aerosol concentrations, while other ice formation processes (homogeneous, deposition freezing, etc.) remain unmodified as in the default SBM and Morr2 microphysics schemes in the WRF v4.2.

To validate the dust simulated by the GEOS-Chem-APM model, the MERRA-2 dust mixing ratio reanalysis is used to qualitatively compare with the dust simulation. Figure 1 a shows the time series of the GEOS-Chem-APM dust number concentration (red dashed line) and PM2.5 dust mixing ratio (red solid line) simulations and MERRA-2 PM2.5 dust mixing ratio reanalysis (blue line) in Jul. 2006. The comparison shows that the time series of the GEOS-Chem dust simulation is generally consistent with the MERRA-2 reanalysis, predicting the strong dust signal from 7–11 July 2006 during the selected severe precipitation event. The vertical distribution of $N_d$ from the GEOS-Chem-APM is also consistent with that of the MERRA-2 reanalysis (Fig. S1), with the highest dust loading at ~1 km and decreasing with altitude. Overall, the GEOS-Chem-APM model captures the occurrence and vertical distribution of this low-level transport of dust, and the dust profile used in the Dusty runs is reasonable to represent the actual dust conditions of this case.

### 2.2.2 Supersaturation-based Condensation

Previous studies indicate that the saturation adjustment approach for condensation and evaporation in the Morr2 microphysics scheme influences the simulations of the invigoration effects of aerosols on the deep convective clouds (Lebo





and Seinfeld, 2011; Lebo et al., 2012; Wang et al., 2013; Khain et al., 2015; Zhang et al., 2021). To figure out the potential influences of the saturation adjustment on the dust-cloud interaction in this case, we modified the Morr2 scheme by replacing

the saturation adjustment assumption with the explicit calculation of condensation and evaporation based on supersaturation. Following Lebo and Seinfeld (2011), the change of the water vapor surplus    within each timestep as a result of phase changes based on the microphysical forcing is expressed as:

$$\frac{d\Delta q_v(t)}{dt} = -C\Delta q_v(t) \tag{1}$$

where $\Delta q_v = q_v - q_{v,s}$, $q_v$ is the water vapor mixing ratio, $q_{v,s}$ is the saturation water vapor mixing ratio. $C$ is a function

of temperature ($T$), pressure ($P$), droplet mass ($q_c$), and number concentrations ($n_c$), with details given in Pruppacher and Klett (1997) and Porz et al. (2018). The supersaturation in Eq. (1) is solved by analytical integration over the time step, and the integrated water vapor surplus change is used to calculate the cloud droplet condensation. The growth equation of $q_c$ following Porz et al. (2018) is expressed as:

$$\frac{dq_c}{dt} = \rho d n_c^{\frac{2}{3}} q_c^{\frac{1}{3}} \Delta q_v \tag{2}$$

where $\rho$ is the air density and $d$ is a function of pressure (P) and temperature (T) defined as:

$$d = 4\pi\left(\frac{3}{4\pi\rho_w}\right)^{\frac{1}{3}} DG \tag{3}$$

$\rho_w$ is the density of liquid water, $D$ is the diffusion term depending on $T$ and $P$, and $G$ is the term of latent heat influence.

In the Morr2-Clean and Morr2-Dusty, Eq. (2) is used to replace the saturation adjustment approach in the Morr2 scheme and integrated analytically over the time step to explicitly calculate condensation/evaporation growth of cloud droplets. These

two simulations are referred to as Morr2-ec-Clean and Morr2-ec-Dusty, respectively. Due to the constant droplet concentration assumption in the Morr2 scheme, the influences of supersaturation on the cloud condensation nuclei activation are not considered in this study.

## 3 Results

### 3.1 Simulated dust effects on precipitation

In order to investigate the influences of dust aerosols on cloud and precipitation, the simulation results of the six runs are compared and evaluated with available observations and reanalysis. Figure 2a shows the 24-h accumulated precipitation ($r_{24h}$) from 0000 to 2300 LT on 08 July from the TCCIP observation system. The observation shows that the rain band is located along the west coast of Taiwan, and two precipitation centers located at ~23.25°N & 120.5 °E and ~22.5 °N & 120.5 °E, respectively, with the heavy rainfall amount larger than 220 mm d$^{-1}$. Overall, the six model runs capture the spatial pattern of

the observed precipitation on the windward slope of the mountain region, consistent with the observation. However, distributions of the rainfall centers simulated by SBM-Clean (Fig. 2c), Morr2-Org (e), and Morr2-Clean (f) runs mismatch


with the rainfall measurement. SBM-Org (b), SBM-Dusty (d), and Morr2-Dusty (g) generally capture the observed double-center rainfall distribution. From Morr2-Clean to Morr2-Dusty, $r_{24h}$ averaged over all rainfall grids ($r_{24h} > 1$ mm d$^{-1}$) and the strong precipitation grids (top 10 percentiles of the $r_{24h}$, 90$^{th}$ to 100$^{th}$) in the heavy precipitation area (High-Pcp) marked by the

red box increased by around 10% (36 to 40 mm d$^{-1}$) and 15% (122 to 140 mm d$^{-1}$), respectively. From SBM-Clean to SBM-Dusty, $r_{24h}$ averaged over all rainfall grids and the strong precipitation grids in the High-Pcp area increased by about 18% (44 to 52 mm d$^{-1}$) and 20% (128 to 153 mm d$^{-1}$), respectively. The results illustrate that both the bulk and bin schemes predict the stronger accumulated precipitations with the dust effect considered, and the SBM scheme simulates the enhancement of the 24-h rainfall in better agreement with the observation. Nevertheless, the WRF model underpredicts the intensity of the rainfall

centers in all six runs to differing degrees. Among them, SBM-Dusty simulates the highest rainfall amount with ~200 mm d$^{-1}$ in the precipitation centers, which is still weaker than the observation. Some factors could lead to this underestimation predicted by the WRF runs. One of the possible reasons is that the contribution of natural and anthropogenic aerosols is not considered in this numerical study, and another factor could be the relatively coarse resolution (3-km), which cannot resolve the orographically forcing and mountain-valley circulation well.

200       To quantitatively exhibit the difference among the six runs, the Taylor diagram (Fig. 3 and Table 2) is used to evaluate the simulations of the 24-h accumulated precipitation. Dusty runs simulate improved precipitation patterns compared to the Org and Clean runs, with spatial correlation coefficients (*r*) improved from ~0.56 to 0.62 and from ~0.55 to 0.66 with Morr2 and SBM schemes, respectively. SBM-Dusty simulates the stronger precipitation more consistent with the observation, with normalized standard deviation increased from ~0.58 to 0.67, and normalized bias decreased from ~−33% to −24.7%,

respectively. Although Morr2-Dusty predicts a better rainfall pattern than the Org and Clean runs, its simulation of the 24-h accumulated precipitation amount shows no apparent improvement.

As shown in Fig. 2, the strong precipitation, based on observation and simulations, is mainly located in the High-Pcp area, which is selected to investigate the precipitation and cloud properties in detail. The hourly rain rates ($r_h$) by the six runs and the IMERG half-hourly observation are averaged among the precipitation grids ($r_h > 0$ mm h$^{-1}$). The 24-h accumulated

precipitation from IMERG is given in Fig. S2, with the intensity and distribution consistent with the rain gauge-radar observation from TCCIP (Fig. 2a). In Fig. 4, the observed hourly rain rate (black line) shows two peaks, the weaker one occurred in the morning (LT 06:00–10:00) (P1, maximum $r_h = 3.5$ mm h$^{-1}$) and the stronger one in the afternoon (LT 11:00–15:00) (P2, maximum $r_h = 7.1$ mm h$^{-1}$). The observed bimodal distribution is generally captured by all six runs. However, the simulated rain rate intensities are generally weaker than the observation, especially during the P2 period. With both Morr2 and

SBM schemes, the Dusty runs show stronger rain rate intensities at the two peak times. During the period of P1, the maximum $r_h$ simulated by SBM-Dusty is 3.1 mm h$^{-1}$, about 45% stronger than SBM-Org and Clean runs; during the P2 period, SBM-Dusty simulates maximum $r_h = 5.2$ mm h$^{-1}$, about 38% and 55% stronger than SBM-Org and Clean runs, respectively. SBM-Dusty also captures the observed precipitation evolution, which is the weaker rainfall intensity of P1 compared with P2. During the P1 period, the maximum rain rate simulated by Morr2-Dusty is 4.2 mm h$^{-1}$, about 24% stronger than Morr2-Org and Clean

runs. During the P2 period, Morr2-Dusty simulates maximum $r_h = 4.0$ mm h$^{-1}$, about 18% and 42% stronger than Morr2-Org



and Clean runs. It is notable that, Morr2-Org and Clean dramatically overestimated the precipitation from LT 1600 to 2000 compared to the observation, and the false signal is reduced when the dust effect is considered (Morr2-Dusty). Although the total rainfall volumes simulated among the Morr2 runs are similar (Fig. 3), Morr2-Dusty predicts an improved precipitation development than the Org and Clean runs with maximum rain rates closer to the observation. Morr2-Dusty still simulates a

stronger rainfall intensity of P1 than P2, mismatching with observed precipitation evolution.

Therefore, connecting dust aerosols with ice nucleation improves the predicted intensity, distribution, and development of precipitation with both the bulk and bin schemes. In this case, the result also suggests that the SBM scheme outperforms Morr2 in simulating the rainfall intensity, distribution, and temporal evolution. The dust effects (Dusty vs. Clean) increase the rainfall intensities in the heavy precipitation area and periods, and SBM simulated a stronger dust effect than Morr2.

**3.2 Dust impacts on cloud properties**

We further investigate the dust effects on the cloud evolution by investigating the time series of the total (ice + liquid) water content (TWC). Figure 5a–d show that WRF runs predict the time series of cloud properties with two distinct periods of convection, explaining the temporal distribution of the hourly rain rates in the observation and simulation. The comparisons between Clean and Dusty runs (Fig. 5a & c vs. 5b & d) suggest significant impacts of dust on cloud development. In the Morr2-

Dusty and SBM-Dusty runs, the high $N_d$ enhances the heterogeneous freezing process (immersion freezing), leading to the increased TWC during the strong convection periods (P1 and P2). From Morr2-Clean to Dusty runs, the maximum ice water content (IWC$_{max}$) increased by about 50% and 35% during P1 and P2, respectively. From SBM-Clean to Dusty runs, the maximum liquid water contents (LWC$_{max}$) increase by ~20% and 30%, and the IWC$_{max}$ increase by about 70% and 60% during P1 and P2, respectively. Along with the enhanced TWC, the mean cloud top height (CTH) during P2 increases from ~12 km

to 14 km with Morr2 and SBM schemes, when the long-range transport of dust is considered.

During the strong convective period (P2) with the observed precipitation peak, the MODIS Aqua satellite passed the simulation region at LT 14:00. To further investigate the dust effects on the simulations of cloud properties, we use the MODIS observations of the cloud water path (CWP, liquid water path + ice water path) and CTH as references to compare with the Morr2 and SBM simulations in Clean and Dusty conditions. This study concentrates on the mixed-phase and convective cloud,

and the grid cells are selected with a minimum TWC of 0.05 g kg$^{-1}$ and minimum CWP of 100g m$^{-2}$ (Choi et al., 2010; Huang et al., 2015). MODIS observations (Fig.6 a & f) show that, during P2, the convective cloud is distributed along the mountain region, especially over the northern part of the island. The observed high values of CWP and CTH along the south-western coast, indicate that the deep convective cloud is concentrated in the key area as given in Fig.2 (120.2°E–120.75°E, 22.25°N–23.25°N). Now we examine the dust effects on the convective cloud simulated with SBM and Morr2 microphysics schemes.

In comparison to the MODIS observations, the Morr2 and SBM runs generally capture the cloud distribution over the mountain region but underestimate the observed cloud coverage and cloud top height (over 15 km). This underestimation could be caused by various factors: one possible reason is the limitation of the minimum TWC cannot be applied in the MODIS observation to





eliminate the high-altitude cirrus cloud; also the underprediction of CTH could be related to the relatively coarse model resolution. The comparisons between Clean (Fig.6 b & d) and Dusty (Fig.6 c & e) runs show that the cloud coverages are

enhanced in the key area when the dust effect is considered. At the same time, Fig.6 g–j show that the dust effect enhances the CTH from 12 km to over 14 km (Fig.6 g & i vs. Fig.6 h & j), more consistent with the MODIS measurement (CTH >15 km).

During the P2 period, the simulated rain rate, cloud coverage, and CTH are notably enhanced due to the dust effects. The change of these cloud properties can be caused by both convective invigoration and the microphysics effect (Futyan and Del Genio, 2007; Fan et al. 2013; Gibbons et al., 2018). Thus, we focus on the strong convective period P2 to further study in detail

the dust effects on cloud properties.

In this numerical study, mineral dust is directly connected with the cloud ice/snow particle formation through the immersion freezing process. The changes of the hydrometeor species are analyzed to exhibit the dust microphysical effects in detail. Figure 7 shows vertical profiles of number concentration and mixing ratio of the hydrometeor particles averaged among the rainfall grids (>0 mm h$^{-1}$) in the High-Pcp area in the P2 period. In Fig. 7 the Morr2 and SBM schemes simulate similar

vertical distributions of cloud hydrometeors, with the largest IWC and LWC appearing at ~8 km and under 3 km, respectively. The comparisons between Clean and Dusty runs (Fig. 7 blue vs. red) show notable dust effects in the mixed-phase layer (~5–11 km). With both Morr2 and SBM schemes, dust aerosol remarkably enhances the formation of snow particles in the mixed-phase layer, with the snow number concentration about doubled and snow mixing ratios increased by ~60–70% at ~8 km altitude. From clean to dusty conditions, the increased snow particles cause the stronger riming and collision-

coalescence processes, leading to the increased number and mass mixing ratios of graupel in both Morr2 and SBM runs.

In Fig.7, from Clean to Dusty conditions, the raindrop number and mass concentrations notably increase in the mixed-phase regime (around 4–9 km) and under the melting layer (under ~5 km altitude) with both Morr2 and SBM schemes. The dust aerosol not only influences the LWC but also shows impacts on the raindrop particle size distribution (PSD), which is an important factor influencing the collisional growth of raindrop and the cloud microphysics precipitation efficiency. The PSD

in the SBM microphysical scheme is represented by the particular bin microphysical information, and the PSD in the Morr2 scheme is represented by gamma functions based on the predicted hydrometeor mass, number, and the fixed shape parameters (Morrison et al., 2009). The particle mass spectrum size distributions of the snow particle in the mixed-phase layer (4–7 km) and raindrop under the melting layer (under 4 km) averaged in P2 are given in Fig. 8. The result shows, during the P2 period, more and larger snow particles are generated through the enhanced immersion freezing process in the dusty condition (Fig.8

a), and the melting of larger snow particle further leads to the increased raindrop size under the melting layer and enhanced surface rain rate.

The more numerous and larger hydrometeor particles generated in the dusty atmosphere could also be a result of the enhanced convective intensities due to the dust effects. As described in Section 2.2.1, $N_d$ is only connected with the immersion freezing processes of the microphysics schemes, the direct radiative forcing by dust aerosols is not considered. The dynamic

and thermodynamic effects of dust are caused through latent heating from the formation and growth of hydrometeor particles. To tease out the dust aerosol impacts on the convection, the vertical profiles of mean updraft velocities and latent heating rates





averaged over the strong convective grids of the top $10^{th}$ percentile updrafts (with the value greater than ~0.5 m s$^{-1}$) during the P2 period are given in Fig. 9. The result shows that the dust effect leads to the enhancement of the rimming latent heating in the mixed-phase layer (~5–10 km) with the Morr2 and SBM schemes (Fig. 9a-b). In this process, with the dust effects

considered, the stronger riming and Wegener-Bergeron-Findeisen (WBF) processes lead to increasing latent heat release in the mixed-phase regime, which enhances the convective intensities. However, the convective invigoration effects lead to different further impacts on the updrafts and condensation and deposition processes in the Morr2 and SBM runs. From Morr2-Clean to Morr2-Dusty runs, dust effects notably increase the condensation latent heating at 4–8 km altitude (Fig. 9a), leading to the significantly increased convective intensity (by ~34%) at the same level (Fig. 9c). At the upper level (8–12 km), the

Morr2 scheme predicts limited dust effects on the deposition latent heating and the convective intensity. With the SBM microphysics scheme, dust aerosol shows a weaker convective invigoration effect at the lower level comparing to that in the Morr2 runs. Meanwhile, at upper levels, the SBM scheme predicts a stronger enhancement of deposition latent heating due to the dust effect, leading to the notably increased convective intensity (increased by ~ 20% at 10 km, and about doubled at higher altitudes) (Fig. 9b & c).

The analysis shows the effects of dust aerosols on cloud and precipitation, enhancing the convective system via microphysical processes and latent heat release. At the same time, the convection system also has impacts on the advection and vertical transport of dust. The $N_d$ vertical profiles (Fig. 1c) show that without the convective transport of dust, the dust loadings above the freezing level (> 6 km) are similar in both Clean and Dusty conditions ($N_d$ < 0.002 cm$^{-3}$). Figure 10 shows that, even with the orographic uplifting effect, the low-level transported dust aerosols are distributed below the freezing level

(0 ºC) without the occurrence of convections, which have no effects on warm clouds and precipitation without ice-phase processes. During the strong convection stage (P1 and P2), dust aerosols are lifted to the upper level and $N_d$ can reach about 0.12 cm$^{-3}$ at 9 km altitude (~ −20 °C), about two orders of magnitude higher than in the clean atmosphere, which could have considerable impacts on the cloud ice formation and enhance the development of cloud and convection in return.

The result above shows the differences in simulated dust effects on the cloud microphysical and convective properties

with the Morr2 and SBM microphysics schemes. The Morr2 microphysics scheme predicts the more notable enhancement of LWC than the SBM scheme in and under the mixed-phase layer; at the upper level (~ 9–12 km), with dust effects considered, the snow and graupel mass concentrations increase more significantly with the SBM scheme than with the Morr2 scheme (Fig. 7 c vs. d). And comparing to the Morr2 scheme, the SBM scheme predicts a weaker increase of updraft at the mid-level (around 4–8 km) and a more dramatic enhancement of updraft at the upper level (~ 8–12 km) (Fig. 9). These different dust effects

could be caused by the different assumptions and parameterizations of the various microphysical processes between bulk and bin microphysics schemes. Among various factors, the saturation adjustment approach in the bulk scheme is indicated to have impacts on the simulations of the aerosol-cloud interaction processes comparing to the explicit representation of the supersaturation-forced condensation growth in the bin models (Lebo and Seinfeld, 2011; Lebo et al. 2012; Khain et al., 2016; Porz et al., 2018; Zhang et al., 2021). As described in section 2.2.2, based on the Morr2-Clean and Morr2-Dusty, the saturation

adjustment approach is replaced with the explicit condensation growth (as Morr2-ec-Clean and Morr2-ec-Dusty).





The comparison shows the differences in latent heating (Fig. 9a vs. Fig. 11a), updraft (Fig. 9c vs. Fig. 11b), and hydrometeor mixing ratio (Fig. 7c vs. Fig. 11c) simulated by the Morr2 and Morr2-ec runs. The results demonstrate the differences in the prediction of dust effects, that explicit supersaturation scheme produces the weaker enhancement of the condensation (increased by ~30% from Morr2-ec-Clean to Morr2-ec-Dusty) than the saturation adjustment approach

(increased by ~60% from Morr2-Clean to Morr2-Dusty) at the mid-level (~ 4–7 km). And at the upper level (~ 8–12 km), Morr2-ec predicts the enhanced deposition latent heating from clean to dusty conditions (Fig. 11a), which is not predicted by the saturation adjustment scheme (Fig. 9a). Accordingly, from Clean to Dusty conditions, Morr2-ec predicts a smaller increase in liquid hydrometeor mixing ratio at low-altitude and a larger enhancement of the ice particle mass concentrations at the upper level (Fig. 11c). The latent heating rate can influence the buoyancy and updraft strength, thus the Morr2-ec predicts a smaller

increase in updraft strength at the mid-level (~ 4–7 km) and more obviously invigorated convection at upper altitude (~ 8–12 km). The possible reason for the differences between the saturation adjustment (Morr2) and explicit supersaturation approach (Morr2-ec) is that the use of saturation adjustment approach turns all excess water vapor into liquid water at the end of each time step, leading to the stronger enhancement of condensation and latent heating at mid-low levels, and limits buoyancy increase and convective invigoration at upper levels, which is demonstrated in previous studies (Lebo et al., 2012; Grabowski

and Morrison, 2017). The different simulations from the Morr2, SBM, and Morr2-ec scheme runs suggest that the use of saturation adjustment approach in the bulk scheme is one of the important factors contributing to the different dust effects predicted by the Morr2 and SBM runs. The accumulated precipitation and rain rate (Fig. S3 & S4) from Morr2-ec runs are still different from the SBM runs, indicating that the saturation adjustment approach cannot explain all the differences in the predicted dust-cloud interactions by the bin and bulk schemes.

## 340   4 Conclusions and Discussion

This study explores the influences of the long-range transported (under ~5 km) mineral dust on the convective cloud and precipitation. We have conducted numerical simulations of a heavy rainfall case occurring over the mountain region in Taiwan, using the WRF model with the Morrison and SBM cloud microphysics schemes. The dust-related immersion freezing parameterization is implemented in the bulk and bin schemes to connect the model with the dust number concentrations in

clean and dusty conditions. Multiple datasets including TCCIP precipitation observation, satellite retrieved measurements, and reanalysis are used as references to compare with the model simulations.

The precipitations simulated by the default and modified schemes are shown to be generally consistent with the observations but underestimate the accumulated precipitation and the peak rain rates. The sensitivity simulation results indicate that the long-range transported dust particles can notably affect the convective cloud and precipitation by acting as ice-

nucleating particles, even with relatively low number concentrations (~0.6 cm$^{-3}$). We find that from clean to dusty conditions, in the heavy precipitation area (High-Pcp), the dust effect enhances the accumulated precipitation by ~15% and ~20% in the precipitation centers and the peak precipitation rate by ~ 41% and ~55% with Morr2 and SBM, respectively. Both Morr2 and





SBM schemes predict improved precipitation patterns (pattern correlations increase from 0.56 to 0.6 with Morr2 and ~0.55 to 0.66 with SBM) and time series (reduce false predicted rainfall) when dust effect is involved. By comparing the precipitations
simulated by the Morr2 and SBM runs, the result also suggests that the SBM microphysics scheme predicts the better rainfall intensity, pattern, and time series than the Morr2 scheme in this case.

During the strong convective period (P2, LT 11:00–15:00), the simulated cloud properties (ice/liquid water contents, cloud area, and cloud top height) are notably impacted due to the dust effects through both convective invigoration and the microphysics effect. The dust aerosols dramatically impact the cloud heterogeneous freezing process during the strong
convective periods, with the ice water content increasing by ~35% and 60% with Morr2 and SBM schemes from clean to dusty conditions. The increased ice and snow particles in the mixed-phase regime cause the stronger riming and WBF processes and enhanced latent heating, which invigorate convective intensity and enhance the condensation and melting, leading to increasing liquid water content and raindrop stronger surface precipitations. During the deep convective period (P2), the mean updraft velocities and latent heating rates averaged over the strong convective grids (over the top 10 percentiles of the updrafts, with
$w > 0.5 \text{ m s}^{-1}$) show that dust aerosol effects remarkably increase the condensation and deposition latent heating rates, leading to the intensified updraft. This study demonstrates the bidirectional influences between dust aerosols and the convective system, i.e., the low-altitude dust aerosols are lifted to the freezing layer by the strong convection, and in return, leading the enhanced convective intensity via microphysical and invigoration effects.

The result also shows that the Morr2 and SBM schemes simulate different invigoration effects of the dust aerosol, with
a much stronger effect with SBM. From Morr2-Clean to Morr2-Dusty runs, the convective intensity significantly increases at 4–8 km altitude (by ~34%) due to the enhanced condensation latent heating. Meanwhile, with the SBM microphysics scheme, dust effects intensify the updraft more significantly at upper levels, corresponding to the increase in high-level latent heating from the deposition process. The different convective invigoration effects of dust simulated by the Morr2 and SBM schemes are indicated partially caused by the saturation adjustment approach used by the bulk scheme. Compared to the explicit
supersaturation forced condensation, the model run with saturation adjustment approach predicts the stronger enhancement of the condensation process at mid-low levels by turning all supersaturated water vapor into liquid water at the end of each timestep, which limits the deposition process and convective invigoration at upper levels.

In the dust-cloud interaction process, it is difficult to untangle the microphysical effect and convective invigoration effect. With the dust effects, the enhanced freezing and riming can increase latent heat release and boost the convection, and an
increase in latent heating can intensify the updraft and increase condensation and deposition processes in turn. In this study, the model simulations (SBM, Morr2, and Morr2-ec) averaged over the rain band area demonstrate that both bin and bulk microphysics schemes can predict the microphysical effects of dust, manifesting as the enhanced immersion freezing and IWC in the mixed-phase layer. But, among the strong convective grids, the different simulations suggest that the dust effect on the intensity and vertical profile of the convection is sensitive to the supersaturation and condensation approaches (Morr2 vs. SBM,
and Morr2 vs. Morr2-ec) in the microphysics schemes. Besides the saturation adjustment, the different dust effects predicted



by the Morr2 and SBM schemes can also be caused by the different assumptions and parameterizations of the various microphysical processes between the bulk and bin microphysics schemes.

This study shows that the dust aerosols transported at low altitudes (below the freezing level) with relatively low number concentrations have remarkable impacts on the orographic convective cloud and precipitation. The extremely low background

dust loading in summer over this region is one possible factor responsible for the significant dust effect, that $N_d$ increases by about two orders of magnitudes from clean to duty conditions. The abundant moisture transported by the circulation could be another influencing factor, which is suggested to be important in the dust-cloud-precipitation interactions by our previous study (Zhang et al., 2020). This study demonstrates that the importance of dust effects in the predictions of convective cloud and heavy precipitation, and the dust effect improves the simulation by the bulk scheme which requires less computational

resources and is widely used in operational weather forecasting. This study could provide a possible pointcut to improve the numerical weather prediction of extreme precipitation events.

***Data availability***. The GEOS-Chem-APM dust number concentration data can be accessed by contacting the corresponding

authors Yanda Zhang (yzhang31@albany.edu) and Fnagqun Yu (fyu@albany.edu). The MERRA-2 inst3_3d_aer_Nv and GPM IMERG half-hourly precipitation data are available at Goddard Earth Sciences Data and Information Services Center (GES DISC) (https://disc.gsfc.nasa.gov/datasets/M2I3NVAER_5.12.4/summary; https://disc.gsfc.nasa.gov/datasets/GPM_3IMERGHHE_06/summary). The TCCIP precipitation data can be applied and acquired from the Taiwan Climate Change Projection Information and Adaptation Knowledge Platform

(http://tccip.ncdr.nat.gov.tw/NCDR/main/index.aspx). MODIS MYD06_L2 data used in this work were acquired from Atmosphere Archive and Distribution System (LAADS) Distributed Active Archive Center (DAAC) (https://ladsweb.modaps.eosdis.nasa.gov/search/order/1/MYD06_L2--61)

***Author contributions***. YZ, FY, and GL developed the project idea. JF contributed the model code. YZ updated the model. YZ

carried out the numerical simulations. YZ and SL analyzed the data, with contributions from JF. YZ wrote the paper with contributions from FY, GL, and JF.

***Competing interests***. The authors declare that they have no conflict of interest.

***Financial support***. This research was supported by the National Science Foundation Partnership for International Research and Education Program (grant no. OISE-1545917) and the National Aeronautics and Space Administration (grant no. NNX17AG35G and 80NSSC19K1275).



***Acknowledgments***. J. F. acknowledges the support of the U.S. Department of Energy Early Career Research Program. The
GEOS-Chem model used in this study is a freely accessible community model managed by the Atmospheric Chemistry
Modeling Group at Harvard University with support from NASA (http://acmg.seas.harvard.edu/geos/).

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







**Figure 1.** (a) Time series of the GEOS-Chem-APM simulated dust number concentration ($N_d$: red dashed line) and PM2.5 dust mixing ratio (red solid line) and MERRA-2 PM2.5 dust mixing ratio reanalysis (blue line) in Jul. 2006, (b) simulation domains and terrain height, and (c) vertical profiles of WRF boundary conditions of dust number concentration in Clean (blue triangles) and Dusty (red dots) runs.




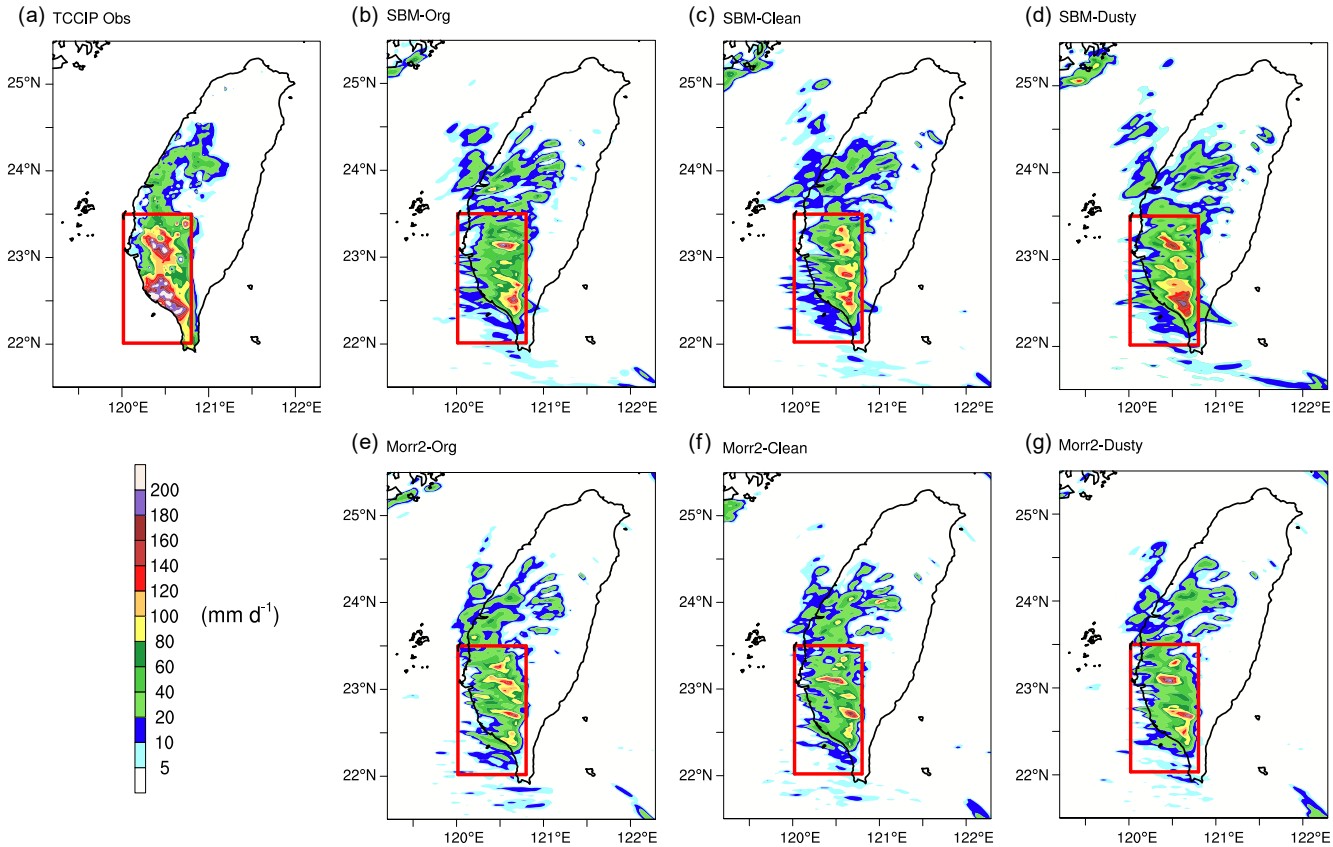

**Figure 2.** 24-hour accumulated precipitation ($r_{24h}$) (LT 2006-07-08) observations from (a) TCCIP, and simulations by (b) SBM-Org, (c) SBM-Clean, (d) SBM-Dusty, (e) Morr2-Org, (f) Morr2-Clean, and (g) Morr2-Dusty. The heavy precipitation is mainly located in the area marked by the red box (High-Pcp).



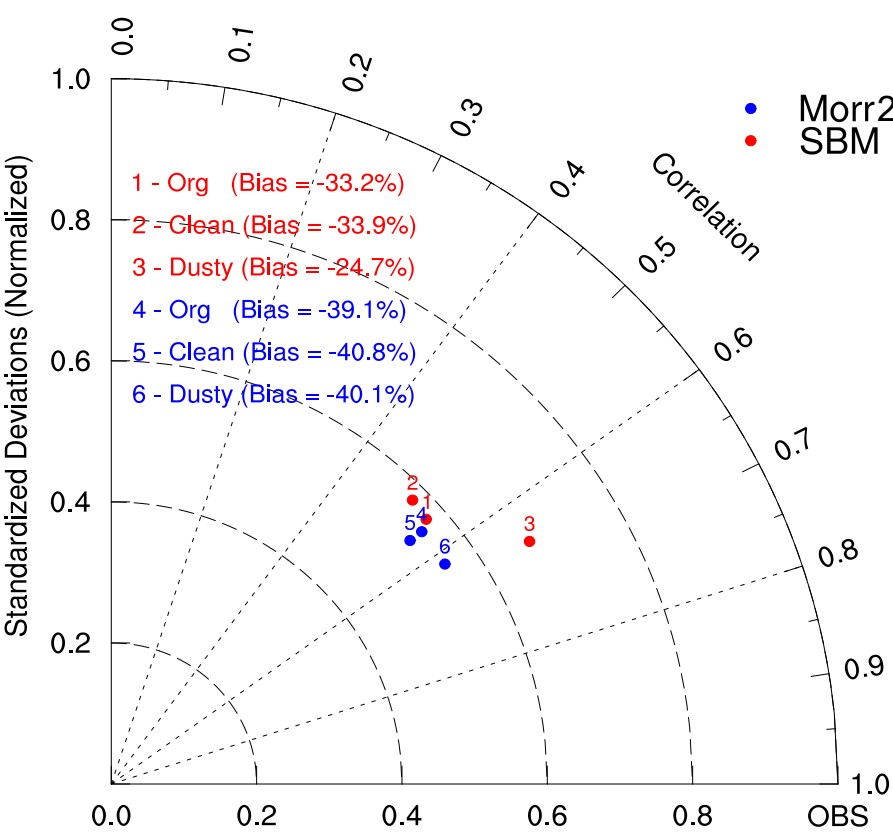

**Figure 3.** Taylor diagram showing spatial correlations coefficients (*r*), standard deviations (SD), and bias (%) between simulations by the six numerical runs and the observation of the 24-h accumulated precipitation.





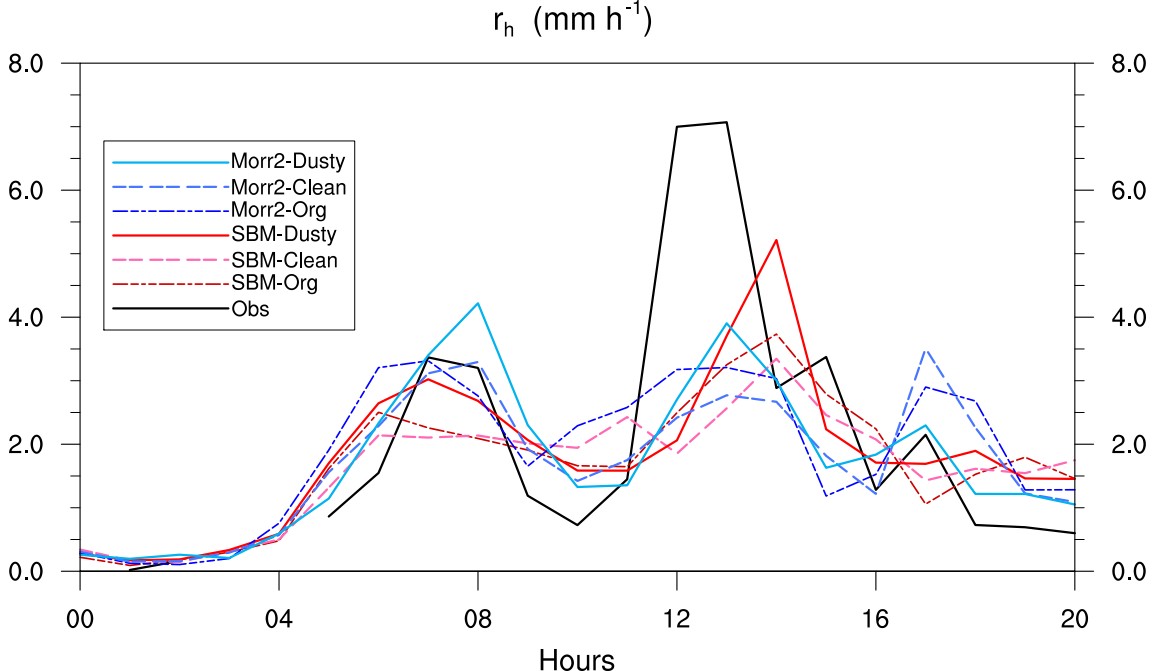

**Figure 4.** Hourly rainfall rates from IMERG observation (black line) and six numerical runs (colored lines) (LT 00:00–20:00, 08 July 2006), averaged among rainfall grids (> 0 mm h$^{-1}$) in the High-Pcp area.





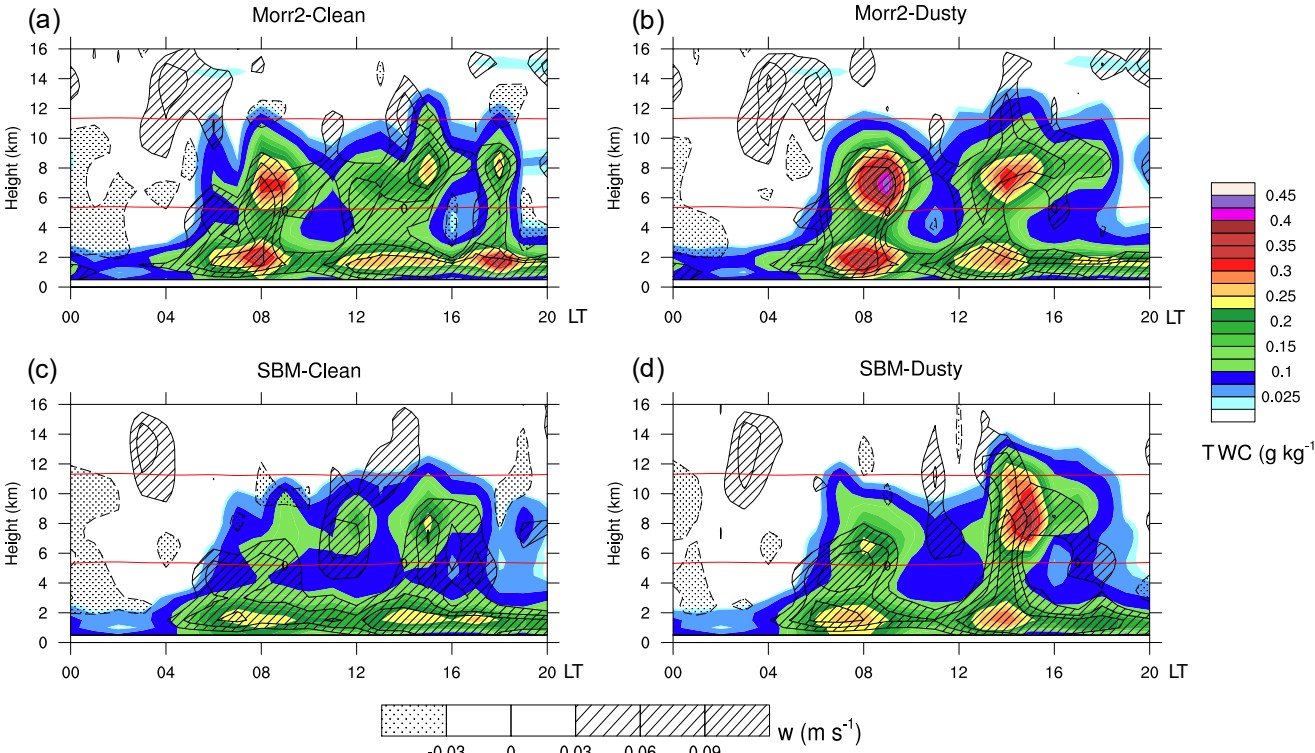

**Figure 5.** (a-d) Time series of total water content TWC (shaded) and vertical velocity (contours) averaged among rainfall grids (> 0 mm h$^{-1}$) in the High-Pcp area for model simulations of Morr2-Clean, Morr2-Dusty, SBM-Clean, and SBM-Dusty, respectively. The two red lines denote the temperature of 0 °C and −38 °C.






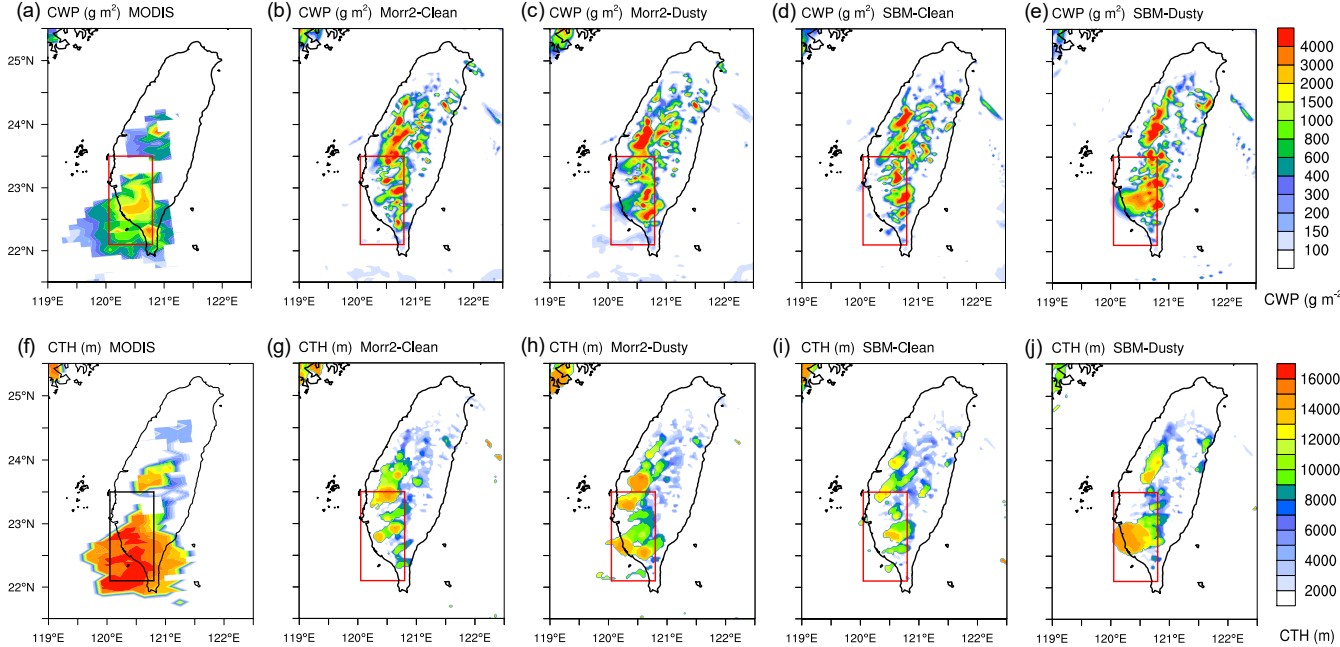

**Figure 6.** The cloud water path (CWP) at LT 14:00 from (a) MODIS observation, (b) Morr2-Clean, (c) Morr2-Dusty, (d) SBM-Clean, (e) SBM-Dusty simulations. Cloud top height (CTH) on grids with CWP larger than 100 g m$^{-2}$ from (f) MODIS observation, (g) Morr2-Clean, (h) Morr2-Dusty, (i) SBM-Clean, and (j) SBM-Dusty simulations.




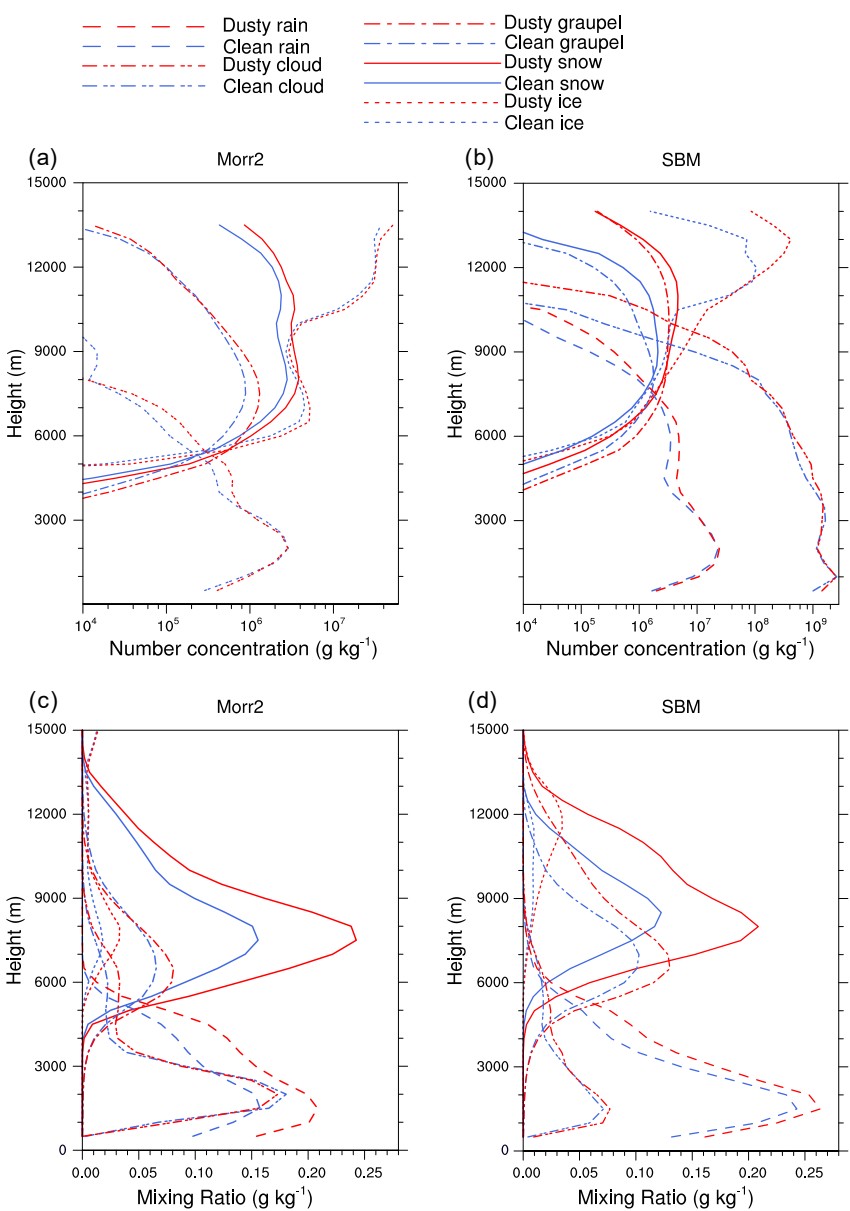

**Figure 7.** Vertical profiles of number concentrations of cloud hydrometers averaged among precipitation grids in the High-Pcp area from LT 11:00–15:00 simulated by (a) Morr2-Clean (blue lines) & Dusty (red lines), and (b) SBM-Clean (blue lines) & Dusty (red lines) runs. Similar time-space averaged vertical profiles of hydrometeor mass mixing ratios by (c) Morr2-Clean (blue lines) & Dusty (red lines), and (d) SBM-Clean (blue lines) & Dusty (red lines).





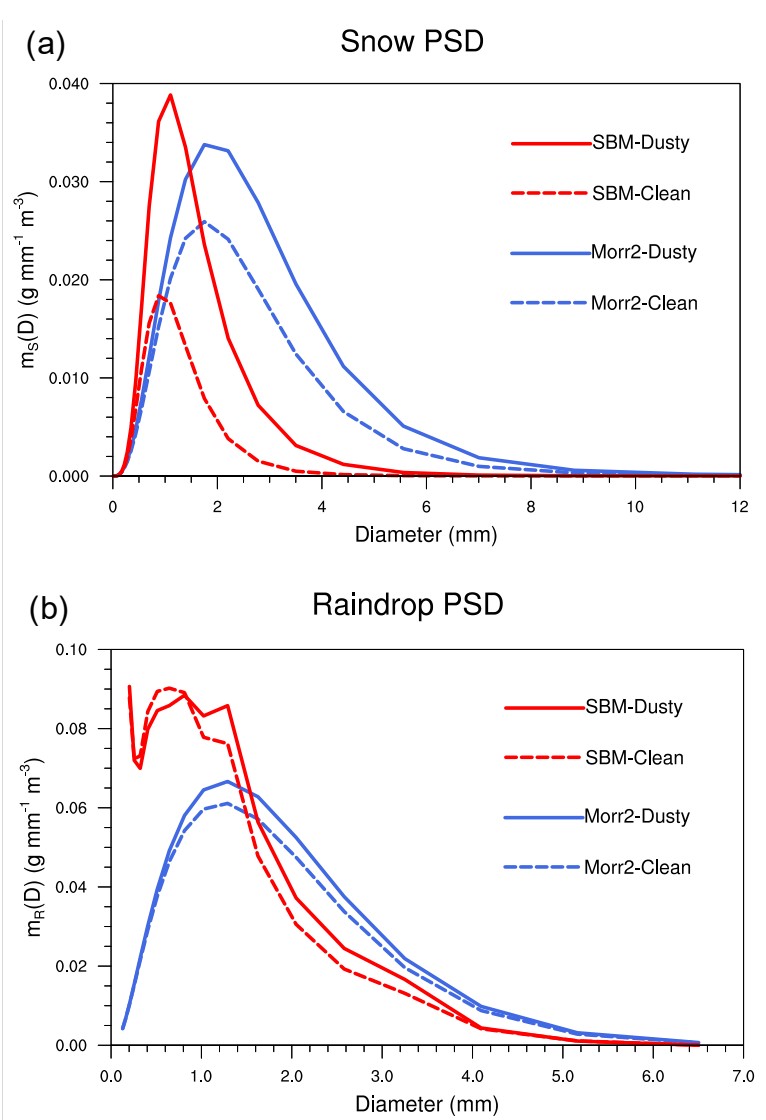

**Figure 8**. Particle mass spectrum size distributions of the (a) snow particle and (b) raindrop from SBM-Clean (red dashed line) & Dusty (red solid line) and Morr2-Clean (blue dashed line) and Dusty (blue solid line) runs averaged over all the precipitation grids in the High-Pcp area during P2.





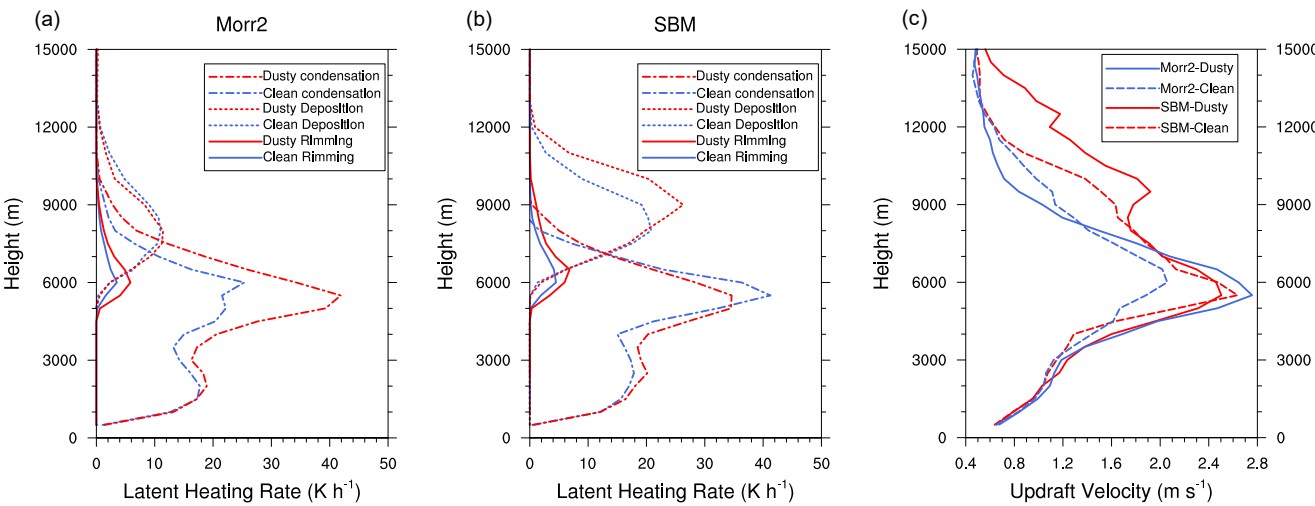

**Figure 9.** (a) Vertical profiles of latent heating rates averaged over the top 10 percentiles of the updrafts ($w > 0.5$ m s$^{-1}$) from
(b) Morr2-Clean (blue lines) & Dusty (red lines), and (c) SBM-Clean (blue lines) & Dusty (red lines) in High-Pcp area during
LT 12:00-16:00 (P2). (c) Similar time-space averaged vertical profiles of updraft velocity averaged over the top 10 percentiles
of the updrafts ($90^{th}$ to $100^{th}$, with w > ~0.5 m s$^{-1}$) simulated by Morr2-Clean & Dusty (blue) and SBM-Clean & Dusty (red).





**Figure 10.** Time series of dust number concentration (shaded) and temperature (red line) and vertical velocity (contours)

averaged among precipitation grids (> 0 mm h⁻¹) in the High-Pcp area from (b) Morr2-Dusty and (c) SBM-Dusty.





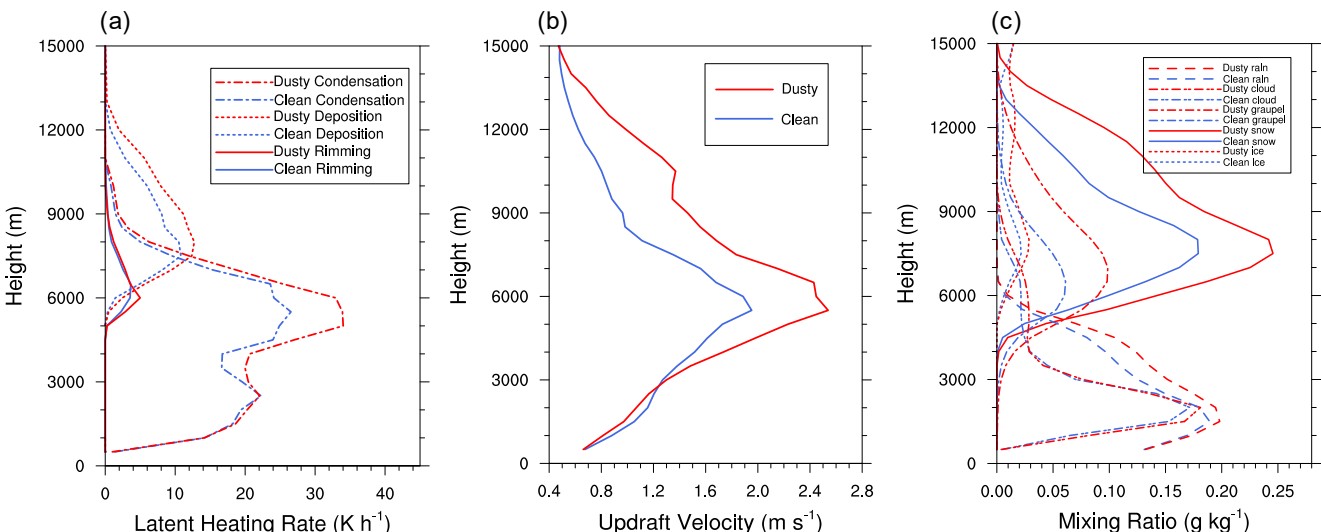

**Figure 11.** Vertical profiles of (a) latent heating rates and (b) vertical velocities averaged over the top 10 percentiles of the updrafts (90th to 100th, with w > ~0.5 m s$^{-1}$) in High-Pcp area during P2 (LT 12:00–16:00) from Morr2-ec-Clean (blue lines) & Morr2-ec-Dusty (red lines). (c) Vertical profiles of cloud hydrometer mixing ratios averaged among precipitation grids in the High-Pcp area during P2 simulated by Morr2-ec-Clean (blue lines) & Morr2-ec-Dusty (red lines).



**Table 1.** Six numerical runs with different settings of the immersion freezing parameterization are considered in this study. The default Morrison and SBM schemes in WRFv4.2.0 with Bigg (1953) immersion freezing parameterization are used as
control runs (Morr2-Org and SBM-Org). Four simulations with modified Morr2 and SBM schemes were run with different dust profiles simulated by the GEOS-Chem model, respectively.

| Microphysics scheme | Experiment runs | Immersion freezing parameterization |
| --- | --- | --- |
| Morr2 | Morr2-Org | Bigg (1953) |
| | Morr2-Clean | DeMott et al. (2015) |
| | Morr2-Dusty | DeMott et al. (2015) |
| SBM | SBM-Org | Bigg (1953) |
| | SBM-Clean | DeMott et al. (2015) |
| | SBM-Dusty | DeMott et al. (2015) |