# Peer review of "Impacts of long-range transported mineral dust on summertime convective cloud and precipitation: a case study over the Taiwan region"

_Atmospheric Chemistry and Physics, 2021_

## Referee Comment (RC1)

Review of "Impacts of long-range transported mineral dust on summertime convective cloud and precipitation: a case study over the Taiwan region" by Yanda Zhang et al.

Recommendation: Minor Revisions

The manuscript "Impacts of long-range transported mineral dust on summertime convective cloud and precipitation: a case study over the Taiwan region" mainly studies a severe precipitation event impacted by dust over Taiwan region in 2006 by using the WRF with properly cloud microphysics parameterizations and immersion freezing parameterization. In general, the paper is well written and presented in a logical way. It is a timely and important piece of work, and of general interest for cloud-aerosol interaction, extreme precipitation and so on. I therefore recommend publication of this paper in Atmospheric Chemistry and Physics after minor revisions. My comments are listed as follows:

Comments:

1. In Figure1a: Dust are usually in coarse mode, and MERRA-2 can provide dust aerosol loadings. Why did authors just use PM2.5? I think it cannot demonstrate the aerosol type.

2. It should be better to add Morr2 and Morr2-ec runs in Table1.

3. There are two peaks of rain rate in this precipitation event. Large part of dust might be rainout during P1, so dust may not invigorate the convection during P2. Authors should consider about that how long the initial dust could affect development of subsequent clouds and precipitation.

4. In Line219: "about 24% stronger than Morr2-Org and Clean runs", is that "24%" for both runs?

5. The aerosol invigoration effect is argued to come from the enhanced latent heating when large amounts of liquid water freeze after being transported above the 0C

level by convective updrafts. It would be much clearer if author mark freezing levels on the latent heating rate profiles in Figure 9 and 11.

6. In Figure 9(b): Why latent heating rate in clean condensation are larger than that in dusty condensation? And is there any mistake in this caption?

7. When talking about long range dust transportation and dust-cloud interactions over East Asia, the authors could cite following references, e.g.

1. Li Z., Y. Wang, J. Guo, et al. 2019: East Asian study of tropospheric aerosols and their impact on regional clouds, precipitation, and climate (EAST-AIR(CPC)). *Journal of Geophysical Research: Atmospheres*. 124 (23), 13026-13054. DOI: 10.1029/2019JD030758.

2. Wang W., J. Huang, P. Minnis, et al. 2010: Dusty cloud properties and radiative forcing over dust source and downwind regions derived from A-Train data during the Pacific Dust Experiment. *Journal of Geophysical Research: Atmospheres*. 115 . DOI:10.1029/2010JD014109.

3. Fu Q., T. Thorsen, J. Su, et al. 2009: Test of Mie-based single-scattering properties of non-spherical dust aerosols in radiative flux calculations. *Journal of Quantitative Spectroscopy & Radiative Transfer*. 110 (14-16), 1640-1653. DOI:10.1016/j.jqsrt.2009.03.010.

---

## Referee Comment (RC2)

In this study, the authors investigate the impact of dust aerosols on convective clouds in summertime over the mountain ranges in Taiwan. The selected case with heavy rainfall occured in July 2006. They used the global WRF model with two different microphysics schemes, the spectral-bin (SBM) and the bulk Morrison (Morr2) scheme. They performed model simulations with and without a coupling of dust to the immersion freezing process using both microphysics schemes to analyze dust effects on convective clouds. The Morr2 scheme uses a saturation adjustment approach, which is replaced in one model simulation by explicit calculations of evaporation and condensation of cloud droplets. The model results are evaluated with observations and the MERRA-2 reanalysis.

I thank the authors for this nice article and recommend it for publication after minor revisions. Please find some general minor comments below and some detailed comments and suggested corrections in the commented manuscript.

- Introduction: You should mention somewhere the typical lifetime of dust particles.

- L45: You mentioned that dust in the mountain ranges of Taiwan originates from different source regions. You could mention that the chemical composition of dust differs for different source regions.

- L65: Reference for long-range transport needed.

- L76: Nd is usually used for the cloud droplet number concentration and it is a bit confusing to use this here for the dust concentration.

- L78: You mean the Nd output are daily and hourly means, right?

- L82: Are there any conclusions about the model performance of the GEOS-Chem model which you can draw from previous studies?

- L137: The long-term mean is for the entire July 2006, right?

- L148: Why do you consider only fine dust PM2.5?

- L175: You mentioned the constant droplet concentration assumption in the Morr2 scheme. How about the SBM scheme?

- L186: Morr2-Clean and Morr2-Dusty look similar in terms of predicting a double-center in the rainfall distribution.

- L197: You mean the CCN activation by aerosols, since the Morr2 scheme has this constant droplet concentration, right?

- L200: Table 2 is missing

[revised manuscript text omitted]

---

## Author Comment (AC1)

MS No.: acp-2021-374

Title: Impacts of long-range transported mineral dust on summertime convective cloud and precipitation: a case study over the Taiwan region

Authors: Yanda Zhang et al. Response to referee comment #1.

We sincerely thank the referee for the detailed reviews and constructive comments which help to improve the manuscript. Below we respond to the comments in detail (*Referee's comments are in Italic*). The manuscript has been revised accordingly.

*Review of "Impacts of long-range transported mineral dust on summertime convective cloud and precipitation: a case study over the Taiwan region" by Yanda Zhang et al.*

*Recommendation: Minor Revisions*

*The manuscript "Impacts of long-range transported mineral dust on summertime convective cloud and precipitation: a case study over the Taiwan region" mainly studies a severe precipitation event impacted by dust over Taiwan region in 2006 by using the WRF with properly cloud microphysics parameterizations and immersion freezing parameterization. In general, the paper is well written and presented in a logical way. It is a timely and important piece of work, and of general interest for cloud-aerosol interaction, extreme precipitation and so on. I therefore recommend publication of this paper in Atmospheric Chemistry and Physics after minor revisions. My comments are listed as follows:*

Many thanks to the reviewer for the comments and suggestions. Please find the point-by-point responses below, and the changes to the manuscript are given by the line numbers of the revised draft with track changes.

*Comments:*

1. *In Figure1a: Dust are usually in coarse mode, and MERRA-2 can provide dust aerosol loadings. Why did authors just use PM2.5? I think it cannot demonstrate the aerosol type.*

Thanks for the comment.

In this study, we use the MERRA-2 reanalysis of dust mass mixing ratio (inst3_3d_aer_Nv) which provides the mass concentration of the dust aerosol in five size bins.

Previous studies suggest that the mass and number concentrations of dust aerosol are generally controlled by particles in coarse and fine modes, respectively (Hoffmann et al., 2008; Kaaden et al., 2009; Mahowald et al., 2014; Denjean et al., 2016). Since this study focuses more on the dust number concentration, we chose the first two bins (diameters 1.46 and 2.8 μm) of the MERRA-2 dust mass mixing ratio to qualitatively compare with the dust mass simulation (diameter from 0.5 to 2.5 μm) from GEOS-Chem-APM.

The Data section is modified to clarify this point (lines 105-110), and adjustments are made accordingly. Related references are added.

2. *It should be better to add Morr2 and Morr2-ec runs in Table1.*

Thanks for the suggestion. The comparisons of Morr2 and Morr2-ec runs are added as Table 2.

3. *There are two peaks of rain rate in this precipitation event. Large part of dust might be rainout during P1, so dust may not invigorate the convection during P2. Authors should consider about that how long the initial dust could affect development of subsequent*

*clouds and precipitation.*

This is a good point. Following the study of Fan et al. (2014), the WRF model is used to study the dust effects, and the dust loss by convective rainout is not considered in this study (as described in Section 2.2.1).

The rainout may impact the dust loading and influence the dust-cloud interactions. However, the convective precipitation in the first case (P1) is a micro-mesoscale system, the long-range transport of dust at a large scale may contribute to dust loading as a supply. Thus we think the dust aerosol should still be able to influence the convection during P2.

In our further study on dust and dust-cloud interactions, the WRF-Chem model will be used, the rainout and dry deposition will be considered as the sink of atmospheric dust.

4. *In Line219: "about 24% stronger than Morr2-Org and Clean runs", is that "24%" for both runs?*

Yes, the "24%" is for both Morr2-Org and Clean runs. During P1, the maximum rain rates simulated by Morr2-Org and Clean runs occur at different times but with similar intensities.

5. *The aerosol invigoration effect is argued to come from the enhanced latent heating when large amounts of liquid water freeze after being transported above the 0C level by convective updrafts. It would be much clearer if author mark freezing levels on the latent heating rate profiles in Figure 9 and 11.*

Thanks for the suggestion. The heights of 0°C (freezing level) and -38°C (homogeneous freezing level) are added in Figs. 7, 9, and 11.

6. *In Figure 9(b): Why latent heating rate in clean condensation are larger than that in dusty condensation? And is there any mistake in this caption?*

We think that this weaker condensation latent heating rate between ~ 5–7 km over the strong convective grids in the SBM-Dusty case could be caused by the dryer updrafts as a result of stronger condensation below the layer (< 5 km) consuming more water vapor (Fig. S5a). The dryer updraft reduces the vapor mixing ratio at ~5–7 km, leading to weaker condensation and latent heating at this level than in the clean condition (Fig. 9b).

This weaker latent heating rate in the dusty condition caused by the dryer updraft is limited within the strong convective grids. Fig. S5b shows that, over the whole High-Pcp area, the dust effect leads to the enhanced latent heating at all altitudes, consistent with the enhanced cloud hydrometers in the dusty condition (Fig. 7).

Fig. S5 give below is added to the Supplement, and the related analysis is added to the manuscript (lines 320–330). The mistakes in the caption of Fig. 9 are revised.

[Figure]

**Figure S5**. In the High-Pcp area during P2: (a) the atmosphere vapor difference (Dusty-Clean) averaged over girds with the top 10 percentiles of the updrafts (90th to 100th) simulated by the SBM runs; (b) vertical profiles of latent heating rates averaged among precipitation grids simulated by SBM-Clean (blue lines) & Dusty (red lines) runs.

7. *When talking about long range dust transportation and dust-cloud interactions over East Asia, the authors could cite following references, e.g.*

*1) Li Z., Y. Wang, J. Guo, et al. 2019: East Asian study of tropospheric aerosols and their impact on regional clouds, precipitation, and climate (EASTAIR(CPC)). Journal of Geophysical Research: Atmospheres. 124 (23), 13026-13054. DOI: 10.1029/2019JD030758.*

*2) Wang W., J. Huang, P. Minnis, et al. 2010: Dusty cloud properties and radiative forcing over dust source and downwind regions derived from A-Train data during the Pacific Dust Experiment. Journal of Geophysical Research: Atmospheres. 115. DOI:10.1029/2010JD014109.*

*3) Fu Q., T. Thorsen, J. Su, et al. 2009: Test of Mie-based single-scattering properties of non-spherical dust aerosols in radiative flux calculations. Journal of Quantitative Spectroscopy & Radiative Transfer. 110 (14-16), 1640-1653. DOI:10.1016/j.jqsrt.2009.03.010.*

Sincere thanks! The recommended papers are now cited in the manuscript (lines 30-50)

**References**

Kaaden, N., Massling, A., Schladitz, A., Müller, T., Kandler, K., Schütz, L., ... & Wiedensohler, A. (2009). State of mixing, shape factor, number size distribution, and hygroscopic growth of the Saharan anthropogenic and mineral dust aerosol at Tinfou, Morocco. Tellus B: Chemical and Physical Meteorology, 61(1), 51-63. https://doi.org/10.1111/j.1600-0889.2008.00388.x

Denjean, C., Cassola, F., Mazzino, A., Triquet, S., Chevaillier, S., Grand, N., Bourrianne, T., Momboisse, G., Sellegri, K., Schwarzenbock, A., Freney, E., Mallet, M., and Formenti, P.: Size distribution and optical properties of mineral dust aerosols transported in the western Mediterranean, Atmos. Chem. Phys., 16, 1081–1104, https://doi.org/10.5194/acp-16-1081-2016, 2016.

Hoffmann, C., Funk, R., Sommer, M., & Li, Y. (2008). Temporal variations in PM10 and particle size distribution during Asian dust storms in Inner Mongolia. Atmospheric Environment, 42(36), 8422-8431. doi:10.1016/j.atmosenv.2008.08.014

Mahowald, N., Albani, S., Kok, J. F., Engelstaeder, S., Scanza, R., Ward, D. S., & Flanner, M. G. (2014). The size distribution of desert dust aerosols and its impact on the Earth system. Aeolian Research, 15, 53-71, doi:10.1016/j.aeolia.2013.09.002

---

## Author Comment (AC2)

MS No.: acp-2021-374

Title: Impacts of long-range transported mineral dust on summertime convective cloud and precipitation: a case study over the Taiwan region

Authors: Yanda Zhang et al. Response to referee comment #2

We sincerely thank the referee for the detailed reviews and constructive comments which help to improve the manuscript. Below we respond to the comments in detail (*Referee's comments are in Italic*). The manuscript has been revised accordingly.

*In this study, the authors investigate the impact of dust aerosols on convective clouds in summertime over the mountain ranges in Taiwan. The selected case with heavy rainfall occurred in July 2006. They used the global WRF model with two different microphysics schemes, the spectral-bin (SBM) and the bulk Morrison (Morr2) scheme. They performed model simulations with and without a coupling of dust to the immersion freezing process using both microphysics schemes to analyze dust effects on convective clouds. The Morr2 scheme uses a saturation adjustment approach, which is replaced in one model simulation by explicit calculations of evaporation and condensation of cloud droplets. The model results are evaluated with observations and the MERRA-2 reanalysis. I thank the authors for this nice article and recommend it for publication after minor revisions. Please find some general minor comments below and some detailed comments and suggested corrections in the commented manuscript.*

Sincere thanks to the reviewer for the comments and suggestions. Please find the point-by-point responses below, and the changes to the manuscript are given by the line numbers in the revised draft with track changes.

*Comments:*

*1.  Introduction: You should mention somewhere the typical lifetime of dust particles.*

Thanks for the comment. The dust aerosol lifetime related content and references are added in the Introduction and related papers are cited (lines 25–26).

*2.  L45: You mentioned that dust in the mountain ranges of Taiwan originates from different source regions. You could mention that the chemical composition of dust differs for different source regions.*

Thanks for the suggestion. Previous studies use multiple methods to study the sources of dust over the Taiwan region, including backward trajectory analyses, satellite observation, site measurement, model simulation, and, as mentioned, the chemical composition of dust. The related description is added (lines 56–58)

*3.  L65: Reference for long-range transport needed.*

As introduced in this paper, previous studies indicate that the atmospheric dust over Taiwan is dominated by the long-range transport of dust and the local dust source is neglectable in most cases. Also, there is no local dust emission over the Taiwan area in the GEOS-Chem model. Thus, we think the dust event on July 8, 2006, is a long-range transport case. We cannot find related references for long-range transport in this specific case. We have slightly modified the text to avoid confusion.

*4.  L76: $N_d$ is usually used for the cloud droplet number concentration and it is a bit*

*confusing to use this here for the dust concentration.*

Thanks for the comments. As this study is closely related to our previous study of the analysis of dust-cloud-precipitation interactions over the Taiwan region (Zhang et al., 2020), we use "$N_d$" to represent the dust number concentration to keep the two papers consistent. We will use other abbreviations to avoid confusing in our future work.

5. *L78: You mean the Nd output are daily and hourly means, right?*

Yes. The daily mean $N_d$ is calculated by averaging the hourly model output, and related adjustments are made (line 87).

6. *L82: Are there any conclusions about the model performance of the GEOS-Chem model which you can draw from previous studies?*

Thanks for the suggestion. In our previous studies on the dust properties over the Taiwan region (Zhang et al., 2019; Zhang et al., 2020a), the comparison between the satellite and surface site observations indicates that the GEOS-Chem model can reasonably capture the occurrence of dust events and variations of dust concentration. The corresponding figure and text are updated in the revision (lines 93–94)

7. *L137: The long-term mean is for the entire July 2006, right?*

Sorry for the unclear statement in the manuscript. Here the "long-term mean" is for all the clean days (with $N_d <$ the 50[th] percentile) in July over the last 30 years (1989-2018). An adjustment has been made to clarify this (line 155).

8. *L148: Why do you consider only fine dust PM2.5?*

This is a good question.

Here we use the PM2.5 dust mass mixing ratio to quantitively evaluated the GEOS-Chem dust simulation, this is because previous studies suggest that the mass and number concentrations of dust aerosol are generally controlled by particles in coarse and fine modes, respectively (Hoffmann et al., 2008; Kaaden et al., 2009; Mahowald et al., 2014; Denjean et al., 2016). As this study concentrates more on the dust number concentration, we chose the first two bins (diameters 1.46 and 2.8 μm) of the MERRA-2 dust mass mixing ratio to qualitatively compare with the dust mass simulation (diameter from 0.5 to 2.5 μm) from GEOS-Chem-APM.

The supplement is added in the Data section to clarify this point (lines 105-110), and adjustments are made accordingly. Related references are added.

9. *L175: You mentioned the constant droplet concentration assumption in the Morr2*

In the SBM microphysics scheme, the lognormal modes are used to represent the concentration, mean radius, and model width of CCN initial and boundary conditions. CCN is activated to be cloud droplets at the cloud base and in the cloud, according to the supersaturation and temperature conditions. This different treatment of cloud droplet is one of many different parameterizations between the Morr2 and SBM schemes, thus is not mentioned in the manuscript.

*10. L186: Morr2-Clean and Morr2-Dusty look similar in terms of predicting a double-center in the rainfall distribution.*

Thanks for the comments. In Fig.2, the simulated precipitation patterns by the Morr2-Clean and Morr2-Dusty runs do look similar. The related contents have been revised in the manuscript (lines 201–205).

*11. L197: You mean the CCN activation by aerosols, since the Morr2 scheme has this constant droplet concentration, right?*

Yes, the cloud droplet number concentration (CDNC) in the Morr2 is constant, and the CCN number concentration in SBM schemes is assumed and described using lognormal modes. Thus, the potential effects of CCN activation by aerosols are not considered in the Morr2 or SBM schemes in this study.

*12. L200: Table 2 is missing.*

Sorry for the mistake. We removed the original "Table 2" for it gave the same information as exhibited in Fig. 3. The correction is made to the draft (around line 220)

**References**

Kaaden, N., Massling, A., Schladitz, A., Müller, T., Kandler, K., Schütz, L., ... & Wiedensohler, A. (2009). State of mixing, shape factor, number size distribution, and hygroscopic growth of the Saharan anthropogenic and mineral dust aerosol at Tinfou, Morocco. Tellus B: Chemical and Physical Meteorology, 61(1), 51-63. https://doi.org/10.1111/j.1600-0889.2008.00388.x

Denjean, C., Cassola, F., Mazzino, A., Triquet, S., Chevaillier, S., Grand, N., Bourrianne, T., Momboisse, G., Sellegri, K., Schwarzenbock, A., Freney, E., Mallet, M., and Formenti, P.: Size distribution and optical properties of mineral dust aerosols transported in the western Mediterranean, Atmos. Chem. Phys., 16, 1081–1104, https://doi.org/10.5194/acp-16-1081-2016, 2016.

Hoffmann, C., Funk, R., Sommer, M., & Li, Y. (2008). Temporal variations in PM10 and particle size distribution during Asian dust storms in Inner Mongolia. Atmospheric Environment, 42(36), 8422-8431. doi:10.1016/j.atmosenv.2008.08.014

Mahowald, N., Albani, S., Kok, J. F., Engelstaeder, S., Scanza, R., Ward, D. S., & Flanner, M. G. (2014). The size distribution of desert dust aerosols and its impact on the Earth system. Aeolian Research, 15, 53-71, doi:10.1016/j.aeolia.2013.09.002